# LongScape: Advancing Long-Horizon Embodied World Models with Context-Aware MoE

## Abstract

Video-based world models hold significant potential for generating high-quality embodied manipulation data. However, current video generation methods struggle to achieve stable long-horizon generation: classical diffusion-based approaches often suffer from temporal inconsistency and visual drift over multiple rollouts, while autoregressive methods tend to compromise on visual detail. To solve this, we introduce LongScape, a hybrid framework that adaptively combines intra-chunk diffusion denoising with inter-chunk autoregressive causal generation. Our core innovation is an action-guided, variable-length chunking mechanism that partitions video based on the semantic context of robotic actions. This ensures each chunk represents a complete, coherent action, enabling the model to flexibly generate diverse dynamics. We further introduce a Context-aware Mixture-of-Experts (CMoE) framework that adaptively activates specialized experts for each chunk during generation, guaranteeing high visual quality and seamless chunk transitions. Extensive experimental results demonstrate that our method achieves stable and consistent long-horizon generation over extended rollouts. Our code is available at: `https://anonymous.4open.science/r/AMSVVD-fdg245`.

## 1 Introduction

Video-based world models have become a prominent research direction in embodied intelligence (Zhu et al., 2024; Liao et al., 2025; Zhen et al., 2025; Shang et al., 2025). By learning environmental dynamics from embodied video data, these models function as powerful synthetic data engines capable of generating large-scale simulated experience to train downstream embodied policy models (such as VLA), thereby alleviating the data scarcity problem in embodied learning (Jiang et al., 2025; Jang et al., 2025; Agarwal et al., 2025). However, current world models are predominantly limited to generating only very short video clips. For more long-horizon embodied tasks in the real world, these models struggle to provide stable generation quality, posing a major challenge to their practical deployment.

Existing video world models can be broadly classified into three categories. The first employs diffusion models (Team et al., 2025; Liao et al., 2025) that apply uniform noise addition and removal across the entire video sequence. Due to the lack of explicit causal structure during training, these models often suffer from error accumulation during rollout inference, resulting in temporal inconsistencies and physically implausible motions over long horizons. The second category adopts autoregressive models (Bruce et al., 2024; Wu et al., 2024) that treat video generation as a next-token prediction task over discretized visual tokens. While these methods preserve temporal causality and support long-term context, they typically yield inferior visual quality compared to diffusion-based approaches. The third category is hybrid models integrating diffusion denoising within an autoregressive framework (Parker-Holder et al., 2024; Deng et al., 2024; Teng et al., 2025; Zhuang et al., 2025). A key limitation of existing hybrid models lies in their use of fixed-length chunks. This rigid chunk partition often cuts a semantically continuous action into separate chunks or combines different action patterns (such as locomotion and manipulation) into a single chunk. As a result, the semantic ambiguity within chunks misguides the next-chunk prediction during autoregressive rollout, leading to inconsistent motions and degraded temporal coherence over long-horizon generation.

To address the challenges of long-term stability in embodied world modeling, we introduce **LongScape**, an embodied world model designed for long-horizon video generation. Our frame-

work adopts a hybrid approach, combining intra-chunk diffusion denoising with inter-chunk autoregressive causal generation. Unlike existing methods that use fixed-length chunks, our key innovation is a context-adaptive, variable-length chunking and generation mechanism. This mirrors how large language models use meaningful tokens, ensuring each of our video chunks represents a complete, semantically coherent action rather than an arbitrary segment. We achieve this by leveraging robot actions as a prior, using the gripper state and the magnitude of effector movement to intelligently determine chunk granularity. Frames with robotic gripper state changes or substantial motion are partitioned into shorter, fine-grained chunks for detailed denoising and high-fidelity generation. Conversely, frames with minimal motion are grouped into longer chunks, which allows for a single, more efficient parallel generation step. We pre-process our embodied dataset using this action-based rule to create video chunks of four distinct granularities. We then train multiple Diffusion Transformer (DiT) experts, with each specializing in a specific video dynamic mode. For instance, experts trained on short chunks are optimized to capture fine-grained object manipulation details, while those trained on long chunks focus on locomotion. To enable adaptive variable-length chunk generation during inference, we designed a context-based dynamic router. At each rollout, this router predicts the expert to be activated by integrating the text instruction and the visual information of the current chunk. This approach ensures that each generated chunk maintains semantic coherence, thereby enhancing causal consistency across the entire generated sequence and leading to more stable long-horizon video generation.

We conduct extensive experiments on the LIBERO and AGIBOT-World benchmarks. Our model achieves state-of-the-art performance in video generation quality compared to existing diffusion, autoregressive, and hybrid baselines. Notably, LongScape can maintain visual coherence and stability over 15 rollouts, demonstrating its long-term generation ability. Our primary contributions are summarized as follows:

- We propose LongScape, a novel embodied world model that conducts adaptive chunk rollout according to the generation context, enabling stable long-horizon video generation.

- We introduce an action-prior-guided chunk partitioning scheme, ensuring the semantic coherence of video chunks, thereby facilitating more effective causal autoregressive generation.

- Extensive experimental results demonstrate the superiority of our model in long-horizon video generation, with sustained stability in visual quality and motion correctness.

## 2 RELATED WORKS

### 2.1 VIDEO GENERATION MODELS

Existing video generation models primarily fall into three architectural categories: diffusion models, autoregressive models, and hybrid diffusion-autoregressive frameworks. Diffusion models learn to transform noise distributions into video data distributions using networks such as UNet (Ronneberger et al., 2015) or DiT (Peebles & Xie, 2023). Early works like VDM Ho et al. (2022) employ 3D-UNet architectures for iterative denoising, while recent open-source models, including CogvideoX (Yang et al., 2024), Hunyuanvideo (Kong et al., 2024), and Wan (Wan et al., 2025), commonly leverage efficient VAEs for spatiotemporal compression and DiT-based denoising to achieve high-quality text-to-video generation. Autoregressive models, inspired by next-token prediction in large language models, tokenize video frames and predict subsequent tokens via transformer architectures. For instance, Cosmos (Agarwal et al., 2025) explores a Llama 3-style autoregressive video generation framework, though such methods still trail diffusion models in visual fidelity. Hybrid approaches seek to combine the benefits of both paradigms: diffusion denoising preserves fine-grained visual quality within chunks, while autoregressive generation ensures temporal coherence and causal consistency across chunks. Representative works like MAGI-1 (Teng et al., 2025) and NOVA (Deng et al., 2024) introduce fixed-length diffusion within an auto-regressive framework to enable chunk-wise text-conditioned and streaming video generation. Our framework advances this hybrid paradigm by introducing adaptive chunk generation, which achieves improved visual quality for long-horizon video generation.

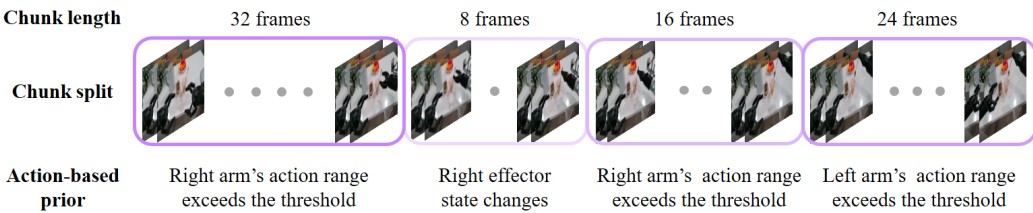

Figure 1: Action-guided chunk partitioning mechanism. This mechanism leverages embodied actions to partition the video, ensuring each chunk contains a distinct semantic unit. It divides the stream into short segments for active manipulation (e.g., significant movement or end-effector state changes) and longer segments for subtle locomotion dynamics.

## 2.2 EMBODIED WORLD MODELS

World models, which leverage generative AI to capture environmental state transitions, hold significant promise for enhancing embodied agents' perceptual understanding and decision-making (Ding et al., 2024). Current approaches primarily follow two distinct paradigms: video generation-based methods and latent space prediction-based methods. The former approach usually adapts open-source video generation foundation models via post-training, incorporating action-conditioned branches to generate future video frames guided by embodied actions (Wu et al., 2024; Zhu et al., 2024; Liao et al., 2025; Huang et al., 2025; Jiang et al., 2025). Recent efforts in this direction (Zhen et al., 2025; Shang et al., 2025) explore integrating multimodal physical cues—such as depth maps, surface normals, and keypoint dynamics—to improve physical realism, spatial accuracy, and motion coherence. Another paradigm advocates for modeling state dynamics in a compact latent space Assran et al. (2025); Baldassarre et al. (2025). For example, V-JEPA 2 (Assran et al., 2025) uses an image encoder pre-trained on large-scale internet data to compress visual inputs into latent representations, followed by an action-conditioned predictor trained to forecast future states. This predictor subsequently facilitates embodied planning via sampling-based optimization. Our work falls within the video world model framework and our key contribution lies in enabling efficient and high-quality long-horizon video generation for embodied scenarios.

## 3 METHODOLOGY

### 3.1 AUTOREGRESSIVE VIDEO GENERATION WITH CHUNK-WISE DIFFUSION

Inspired by the next-token-prediction autoregressive paradigm of LLMs, we approach video generation by treating a video as a sequence of temporal chunks, with each chunk (single or multiple frames) functioning as a "token". This enables a token-by-token rollout to generate videos, a causal paradigm that inherently preserves causality and helps enforce physical consistency. The full video can be represented as $\mathbf{V} = (\mathbf{S}_1, \mathbf{S}_2, \ldots, \mathbf{S}_N)$, where $\mathbf{S}_t$ is the video chunk with a timestamp $t$. The objective is to predict the next chunk based on the preceding history:

$$p(\mathbf{V}) = p(\mathbf{S}_1) \prod_{t=1}^{N-1} p(\mathbf{S}_{t+1}|\mathbf{S}_1, \ldots, \mathbf{S}_t), \tag{1}$$

where $\mathbf{S}_t$ is the video chunk at time step $t$. The central task then becomes modeling the single-step generation process $p(\mathbf{S}_{t+1}|\mathbf{S}_1, \ldots, \mathbf{S}_t)$. Two primary paradigms exist to address this.

The first is to handle visual information similarly to language, using a tokenizer to discretize the image into patches and then extracting discrete representations. These are then arranged in spatiotemporal order to form the chunk's overall token: $\mathbf{X}_t$, where $\mathbf{X}_t = \text{Tokenizer}(\mathbf{S}_t)$. A transformer-based model is then trained on a next-token prediction task to predict the subsequent chunk token $\mathbf{X}_{t+1}$. While this approach is generally efficient, the discretization of continuous visual information can lead to a reduction in generated image quality.

The second paradigm is diffusion-based continuous generation, where the next video chunk, $\mathbf{S}_{t+1}$, is produced through a conditional denoising process guided by the previous chunk, $\mathbf{S}_t$. Operating

within a continuous latent space (often from a VAE), the model starts with a noisy latent representation of the target chunk, $\mathbf{z}_{t+1,K}$. This representation is then iteratively refined over $K$ steps. At each denoising step $k$, a model, typically a U-Net or a Diffusion Transformer (DiT), predicts and removes noise from the current noisy sample, $\mathbf{z}_{t+1,k}$. This operation is conditioned on the preceding chunk $\mathbf{S}_t$, following the process: $\mathbf{z}_{t+1,k-1} = \text{Denoise}(\mathbf{z}_{t+1,k}, \mathbf{S}_t)$. This refinement ultimately yields a clean latent representation, $\mathbf{z}_{t+1,0}$, which is then decoded into the final video chunk, $\mathbf{S}_{t+1}$. This approach preserves finer local visual details, mitigates error accumulation and drift during long-range autoregressive rollout, and benefits from the powerful generative capabilities of modern full-sequence diffusion models. Given these advantages, we adopt a hybrid framework that combines local chunk-based diffusion denoising with global autoregressive generation. In the following section, we elaborate on our design of adaptive chunk partitioning to enable more efficient long-horizon video generation within this framework.

### 3.2 ACTION-GUIDED VIDEO CHUNK PARTITION

Our design for the Mixture of Experts (MoE) in embodied video generation stems from the observation that different phases of a robotic task involve distinct dynamic patterns. For instance, long-range navigation toward an object requires the model to capture locomotion patterns, while precise manipulation demands an understanding of fine-grained dynamics. Therefore, we propose to partition the video sequences according to these different action types.

We posit that robotic action labels (gripper state and position) provide a reliable signal for our video chunk partitioning. Our approach is built upon two factors: (1) changes in the gripper state can indicate whether the robot is interacting with an object, and (2) the magnitude of motion can be quantified using the positional information (i.e., $x, y, z$, pitch, yaw, roll). The video is first divided into non-overlapping base chunks of 8 frames each. A final chunk is defined as a contiguous sequence of 1 to 4 base chunks, corresponding to 8, 16, 24, and 32 frames. To determine the appropriate chunk length, we begin by computing the motion amplitude for each dimension of the positional information across the entire video. A motion threshold $\theta_d$ for dimension $d$ is set as a fraction $\alpha$ (where $0 < \alpha < 1$) of its global amplitude. A chunk partition example is presented in Figure 1.

The partitioning algorithm processes the video sequentially from the initial frame. Starting with the first unassigned base chunk, it iteratively evaluates a candidate chunk consisting of the next $n$ consecutive base chunks ($1 \leq n \leq 4$). The partitioning steps proceed as follows:

1. **Substantial Motion Detection:** If the range of motion in any dimension within the candidate chunk exceeds $\theta_d$, all $n$ base chunks are grouped into a single chunk.

2. **Maximum Length Check:** If $n = 4$, the candidate chunk is immediately assigned as a chunk to avoid exceeding the maximum allowed length.

3. **Gripper State Change Detection:** If the gripper state changes in the last base chunk of the candidate chunk, the first $n-1$ base chunks form one chunk, and the last base chunk is treated as a separate chunk.

4. **Continue Expansion:** If none of the above conditions are met, $n$ is incremented by 1, and the process repeats—unless $n$ already equals 4.

This approach ensures that video segments with significant motion or critical gripper state changes are assigned shorter chunks, while segments with less dynamic activity are assigned longer chunks. The overall process is formalized in Algorithm 1.

### 3.3 CONTEXT-AWARE MoE FOR ADAPTIVE ROLLOUT

Following the partitioning of video into chunks of varying lengths, we introduce a **Context-aware Mixture-of-Experts (CMoE)** framework to reconcile specialized DiT networks for denoising each chunk type. This approach mitigates conflicts and forgetting issues that arise from training on diverse motion patterns, e.g., large-scale locomotion versus fine-grained object manipulation. Our model employs a set of $K$ DiT experts, denoted as $\mathcal{E} = \{\mathcal{E}_1, \mathcal{E}_2, \ldots, \mathcal{E}_K\}$. Each expert is initialized from the parameters of a pre-trained CogVideoX (Yang et al., 2024). In our implementation, we set $K = 4$, corresponding to four distinct chunk types. Each expert network is composed of a stack of transformer blocks, featuring 3D full attention layers and feed-forward networks connected by

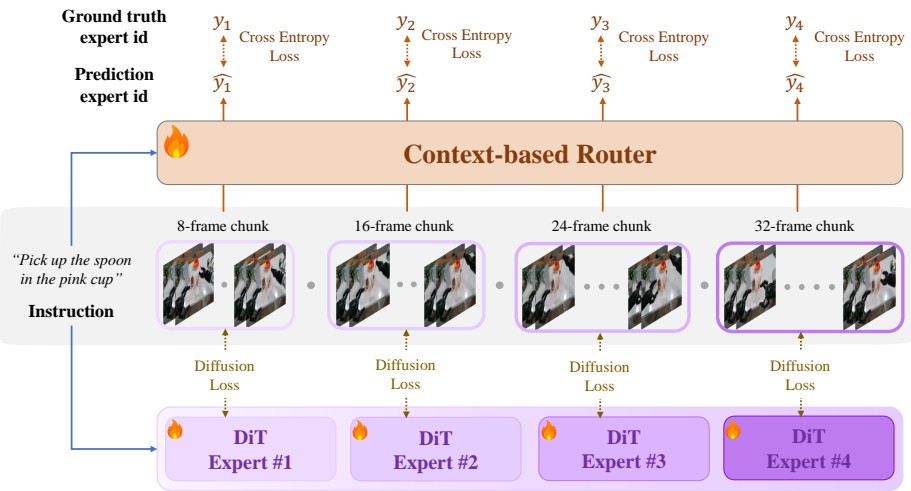

Figure 2: Training paradigm of LongScape. Different types of video chunks are used to train specialized DiT experts, while a context-based dynamic router learns to allocate the next chunk to the appropriate expert based on textual instruction and visual context.

Adaptive Layer Normalization (AdaLN) layers. We train each expert $\mathcal{E}_i$ to perform a denoising task on its corresponding chunk type, optimizing the objective function:

$$\mathcal{L}_i = \mathbb{E}_{\mathbf{z}_0, \epsilon \sim \mathcal{N}(0, \mathbf{I}), \tau \sim [1, T]} \left[ \| \epsilon - \mathcal{E}_i(\mathbf{z}_\tau, \tau, \mathbf{c}, \mathbf{S}_t) \|^2 \right], \quad (2)$$

where $\mathbf{z}_0$ is the clean latent representation of the target chunk $\mathbf{S}_{t+1}$, $\epsilon$ is the added noise, $\mathbf{z}_\tau$ is the noisy latent at diffusion time step $\tau$, $T$ is the total number of diffusion steps, $\mathbf{c}$ is the global text condition, and $\mathbf{S}_t$ is the preceding video chunk. The expert model $\mathcal{E}_i$ predicts the noise $\epsilon$ to be removed, conditioned on the text prompt and the preceding visual context.

To enable adaptive inference-time expert selection, we introduce a **dynamic contextual router** $\mathcal{R}$, which is implemented by a single cross-attention transformer network, taking the global text instruction $\mathbf{c}$ and visual features from the current video chunk $\mathbf{S}_t$ as input, predicting which expert to activate for the next chunk $\mathbf{S}_{t+1}$. The router is trained using a cross-entropy loss:

$$\mathcal{L}_{\text{router}} = \mathbb{E}_{(\mathbf{c}, \mathbf{S}_t, i)} \left[ -\log p(i | \mathbf{c}, \mathbf{S}_t) \right], \quad (3)$$

where $i$ is the ground-truth expert index. The training process of experts and the router is presented in Figure 2 and this process is highly efficient, requiring only a small subset of partitioned video chunks and their corresponding chunk length labels.

The inference pipeline is illustrated in Figure 3. At each rollout step, the global text prompt and visual features from the last frame of the current chunk are fed into the router to predict the appropriate expert:

$$i^* = \underset{i \in \{1, \ldots, K\}}{\arg \max} \left( \mathcal{R}(\mathbf{c}, \mathbf{S}_t)_i \right). \quad (4)$$

The activated expert $\mathcal{E}_{i^*}$ then generates the next chunk of the corresponding length $\mathbf{S}_{t+1}$:

$$\mathbf{S}_{t+1} = \mathcal{E}_{i^*}(\mathbf{S}_t, \mathbf{c}). \quad (5)$$

The MoE mechanism allows us to scale model capacity by a factor of $K$ without significantly increasing the memory footprint during inference, as only one expert is activated at a time.

## 4 EXPERIMENTS

### 4.1 EXPERIMENTAL SETUP

**Datasets.** We conduct experiments on LIBERO and AGIBOT-World datasets, both focusing on long-horizon composite embodied manipulation tasks (e.g., folding clothes).

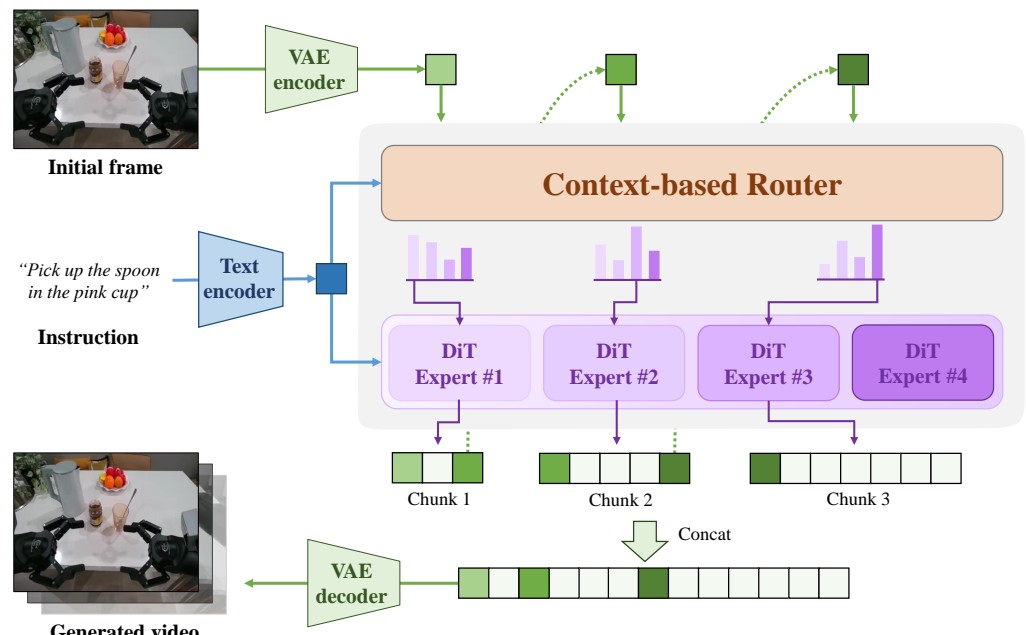

Figure 3: Inference pipeline of LongScape. At each autoregressive step, the dynamic router leverages the global text instruction and the visual features of the current chunk to select the appropriate DiT expert. The selected expert then generates the subsequent video chunk via diffusion denoising, with this iterative process continuing to produce a long video sequence.

- **LIBERO** (Liu et al., 2023): A benchmark for lifelong robot learning, including 130 diverse manipulation tasks organized into specialized suites (LIBERO-Spatial, LIBERO-Object, LIBERO-Goal, and LIBERO-100) to study knowledge transfer across variations in object spatial relationships, object types, and task goals.
- **AGIBOT-World** (Bu et al., 2025): A large-scale real-world dataset collected with a wheeled dual-arm robot. It covers over 100 real-world scenarios across five core environments (household, catering, industrial, commercial, office) and contains a high proportion of long-horizon tasks.

**Baselines.** We compare our approach with three advanced video generation world models, covering purely diffusion-based (CogVideoX (Yang et al., 2024)), autoregressive-based (Genie (Bruce et al., 2024)), and diffusion-autoregressive (NOVA (Deng et al., 2024)) architectures. The details are presented in Appendix A.3.

**Metrics.** We evaluate the generated videos using the following metrics:

- **PSNR**: It evaluates the pixel-level similarity between generated and ground-truth frames.
- **LPIPS**: It measures the image feature similarity of generated and ground truth frames.
- **SSIM**: It assesses the brightness, contrast, and structural consistency between generated and ground truth frames.
- **FVD**: It measures the distance between real and generated video feature distributions.

**Implementation of LongScape.** For the AGIBOT-World dataset, each DiT expert has 5.57G parameters, and the router has 108.57M parameters. For the LIBERO dataset, each DiT expert also has 5.57G parameters, and the router has 71.35M parameters. For data preprocessing, we sampled videos at 10 Hz from both the LIBERO and AGIBOT-World datasets, resulting in approximately 30,000 training clips from each source. Each expert was trained for two epochs, a process requiring roughly 24 hours on four NVIDIA H20 GPUs. The training of the router took around 24 hours on 4 NVIDIA H20 GPUs.

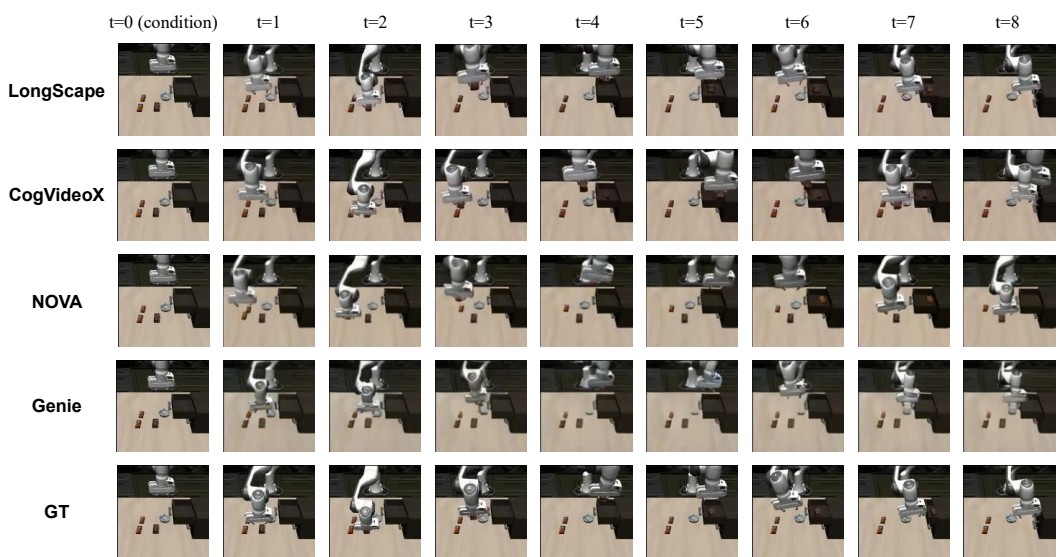

Figure 4: Qualitative comparison on LIBERO dataset with text instruction "put the chocolate pudding in the top drawer of the cabinet and close it". Here we sample eight frames from the generated long video sequences for better presentation.

## 4.2 MAIN RESULTS

**Quantitative comparison.** Quantitative evaluations in Table 1 demonstrate that our model achieves superior performance across all four metrics on both the LIBERO and AGIBOT-World datasets. These benchmarks focus on **long-horizon composite embodied manipulation tasks** (each task lasting approximately 20 seconds), which generally require multiple rollout steps to generate the full video sequence. Compared with the most competitive baseline, LongScape achieves an average 8.6% improvement on LIBERO dataset and 5.8% improvement on AGIBOT-World dataset. For diffusion-based models such as CogVideoX, we generate the entire long frame sequence in a single forward pass. For autoregressive and hybrid models, long sequences were produced through iterative rollouts. However, these methods have significant drawbacks. Diffusion models often generate physically implausible content due to their parallel frame generation mechanism, while the autoregressive models suffer from error accumulation, where early inaccuracies propagate and amplify over time. These limitations hinder both baseline types from producing coherent long-range videos. Our model's architecture avoids these issues by flexibly generating chunks with full semantic integrity. This approach ensures high-quality generation within each chunk and maintains strong causal relationships between them, making our model exceptionally well-suited for long-horizon generation tasks.

**Qualitative results.** To provide a more intuitive comparison of our method's effectiveness against the baselines, we offer a set of visualizations of the generated results. Figure 4 shows the generation of a complex manipulation task from the LIBERO dataset. This task involves multiple steps, such as picking and placing objects and closing a drawer. For autoregressive models, this requires up to 10 rollouts (generating approximately 200 frames in total), demanding a high degree of stability for long-horizon generation. As shown in the visualization, CogVideoX suffers from ghosting artifacts in later frames. NOVA misinterprets the instruction, picking up the wrong object, while Genie fails in the very first pickup action. In contrast, our model maintains coherent actions across multiple rollouts, producing a logical and successfully executed task sequence. Figure 5 presents a challenging cloth-folding task from the AGIBOT-World dataset. This task requires a high level of spatiotemporal coherence and a strong understanding of physical laws. Here, CogVideoX again produces blurry artifacts. NOVA suffers from objects disappearing from the scene. By comparsion, our model consistently maintains a plausible deformation of the cloth and generates a continuous, coherent motion sequence even with multiple rollouts.

Table 1: Quantitative comparison of different video world models in terms of video visual quality on LIBERO and AGIBOT-World datasets.

| Method | LIBERO | | | | AGIBOT-World | | | |
|---|---|---|---|---|---|---|---|---|
| | PSNR↑ | LPIPS↓ | SSIM↑ | FVD↓ | PSNR↑ | LPIPS↓ | SSIM↑ | FVD↓ |
| CogVideoX | 19.315 | 0.1402 | 0.7729 | 184.69 | 15.800 | 0.4028 | 0.6733 | 267.39 |
| NOVA | 13.007 | 0.4488 | 0.4262 | 346.91 | 16.043 | 0.4201 | 0.6998 | 273.60 |
| Genie | 19.300 | 0.2192 | 0.7267 | 383.82 | 16.166 | 0.3620 | 0.6058 | 495.93 |
| **LongScape** | **19.977** | **0.1231** | **0.7883** | **153.72** | **16.493** | **0.3613** | **0.7015** | **256.16** |

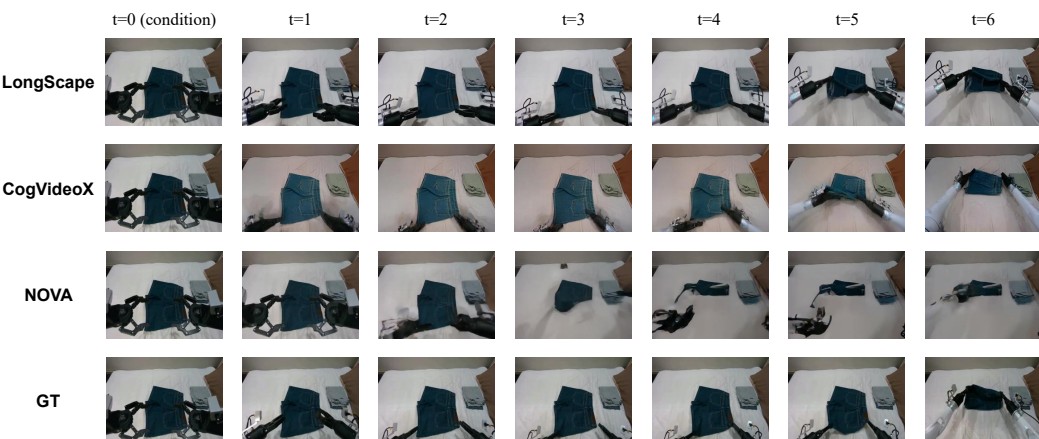

Figure 5: Qualitative comparison on AGIBOT-World dataset with text instruction "grasp the lower pant leg and waistband of denim shorts and fold them over the upper pant leg and waistband". Here we sample six frames from the generated long video sequences for better presentation.

## 4.3 ABLATION STUDY

**Effectiveness of the context-based MoE.** The core of our method is a MoE design, which flexibly handles chunk generation for different contexts, ensuring both intra-chunk coherence and semantic integrity. To validate this design, we conducted an ablation study comparing our model with several variants that use a single expert trained on fixed-length chunks. We tested four such variants, each trained exclusively on chunks of 8, 16, 24, or 32 frames.

As shown in Table 2, the quantitative results on both the LIBERO and AGIBOT-World datasets indicate a performance drop when using a fixed-length expert. Notably, models trained on longer fixed chunks (e.g., 32 frames) performed slightly better than those trained on the shorter one (e.g., 8 frames). We attribute this to the fact that very short chunks fail to capture a complete, semantically meaningful action, making it difficult for the model to learn meaningful motion dynamics.

We also conduct some qualitative analysis, shown in Figure 6 and Figure 8, which further demonstrates the limitations of fixed-length chunk designs. We observed that models using excessively long chunks tend to generate only long-range movements without fine-grained manipulation, while models using very short chunks exhibit poor dynamic quality. This clearly shows that long and short chunks capture different motion patterns: long chunks are suited for global movement, while short ones capture local manipulation details. Our MoE approach, by contrast, flexibly handles these diverse motion patterns, leading to consistently coherent generation.

**Effectiveness of the dynamic router.** To evaluate our dynamic router's ability to reliably predict expert assignments across different scenarios, we framed the task as a four-class classification problem. We tested the router on a test set of 12,700 samples on LIBERO dataset. The router correctly predicted the expert for 11,606 samples, resulting in an accuracy of 91.4%, with a distribution of

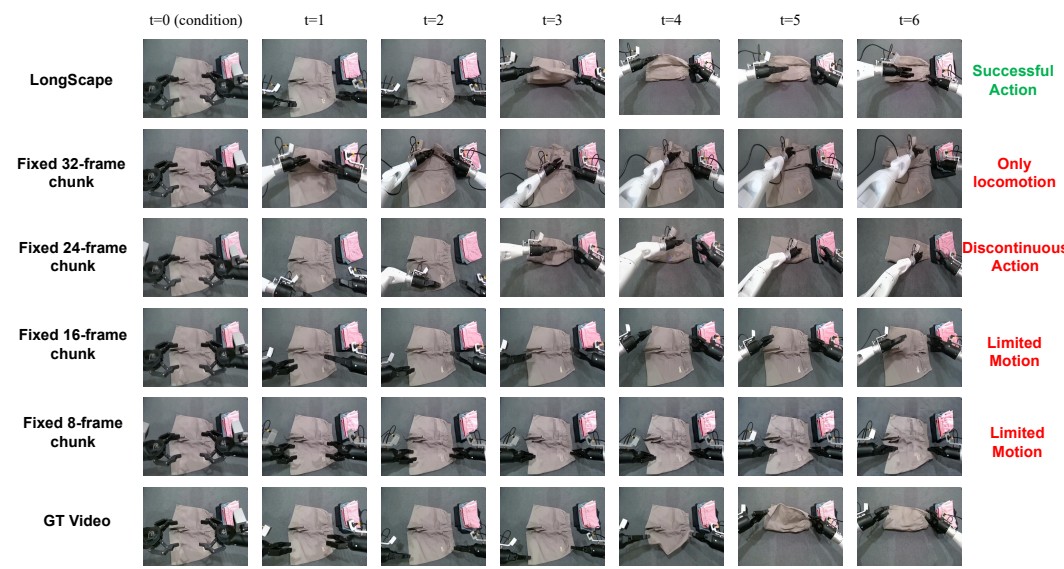

Figure 6: Visualization of the ablation study on AGIBOT-World dataset. We compare LongScape with variants using only fixed-length chunks. These fixed-length approaches often exhibit discontinuous or physically implausible actions. In contrast, LongScape, by incorporating an adaptive expert mechanism, ensures action coherence and stable generation.

Table 2: Ablation study of using various fixed-length chunk generation and the MoE designs.

| Method | LIBERO | | | | AGIBOT-World | | | |
|---|---|---|---|---|---|---|---|---|
| | PSNR↑ | LPIPS↓ | SSIM↑ | FVD↓ | PSNR↑ | LPIPS↓ | SSIM↑ | FVD↓ |
| **LongScape** | **19.977** | **0.1231** | **0.7883** | **153.72** | **16.493** | **0.3613** | **0.7015** | **256.16** |
| 32-frame chunk | 19.755 | 0.1285 | 0.7845 | 180.31 | 15.641 | 0.4286 | 0.6702 | 282.38 |
| 24-frame chunk | 19.620 | 0.1314 | 0.7798 | 191.02 | 15.903 | 0.4034 | 0.6814 | 303.75 |
| 16-frame chunk | 19.640 | 0.1388 | 0.7783 | 176.55 | 15.894 | 0.3852 | 0.6917 | 373.50 |
| 8-frame chunk | 18.893 | 0.1629 | 0.7518 | 229.71 | 15.578 | 0.4052 | 0.6669 | 387.49 |

prediction results shown in Figure 7. This result demonstrates the router's effectiveness in reliably selecting the appropriate expert for a given context.

## 5 CONCLUSION AND FUTURE WORKS

In this work, we introduce LongScape, a novel framework for long-horizon embodied world modeling that adaptively integrates diffusion denoising with autoregressive generation. Our method leverages action priors to guide video chunk partitioning and employs a dynamic router to activate specialized generation experts. This approach enables the production of video tailored to the semantic context of each chunk, significantly enhancing visual quality and coherence. Experimental results demonstrate that LongScape outperforms existing approaches by maintaining stable and coherent video generation over extended horizons.

Despite these advancements, several directions merit further exploration. First, while the current action-guided chunking mechanism relies on heuristic rules, future work could investigate learning-based chunking strategies to achieve more flexible and task-aware video partitioning. Second, extending LongScape to support multi-view camera inputs and generation at the scale of several minutes would further enhance its applicability in complex real-world scenarios.

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

# A APPENDIX

## A.1 THE USE OF LARGE LANGUAGE MODELS (LLMS)

We acknowledge the use of a large language model (LLM) only for assistance with literature search and retrieval.

## A.2 REPRODUCIBILITY STATEMENT

To ensure reproducibility, we have made our code and some visualization cases available in the following anonymous repository for review: `https://anonymous.4open.science/r/AMSVVD-fdg245.`.

## A.3 DETAILS OF BASELINES

We compare our approach with three advanced video generation world models, covering purely diffusion-based, autoregressive, and combined diffusion-autoregressive architectures, which are detailed as follows.

- **CogVideoX** (Yang et al., 2024): This is an advanced open-source diffusion-based video generation model. It introduces 3D full attention to effectively model spatiotemporal relationships. We use the 5B version for our implementation.
- **Genie** (Bruce et al., 2024): This is an autoregressive video generation model that encodes video frames into discrete tokens and employs a spatial-temporal attention-based transformer for video prediction.
- **NOVA** (Deng et al., 2024): This is a recently proposed video generation model that combines a diffusion model with an autoregressive framework. Each frame is generated through diffusion denoising, while inter-frame relationships are established autoregressively to produce a continuous video rollout.

---

**Algorithm 1** Algorithm of chunk partition

---

**Input:** Video frames $F$, action sequence $A$ for each frame
**Output:** Partitioned chunks set $S = \{S_1, S_2, \ldots, S_k\}$

1: Divide video into base chunks $C$ of 8 frames each
2: Calculate action thresholds $\theta_d$ for each dimension $d \in \{x, y, z, \text{pitch}, \text{yaw}, \text{roll}\}$
3: **while** not all base chunks processed **do**:
4:     Check candidate chunk $C_{1:n}$ of length n (1-4 base chunks):
5:     **if** $\Delta A_{d,1:n} > \theta_d$ **then**: add $C_{1:n}$ to $S$
6:     **else if** n=4 **then**: add $C_{1:n}$ to $S$
7:     **else if** gripper state changes in $C_n$ **then**: add $C_{1:n-1}$ and $C_n$ to $S$
8:     **else**: increase n and continue checking

---

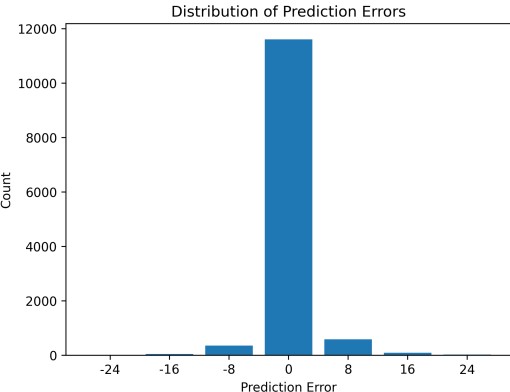

Figure 7: Prediction error distribution of the dynamic router. The router achieved a 91.4% accuracy rate, demonstrating its effectiveness in expert selection.

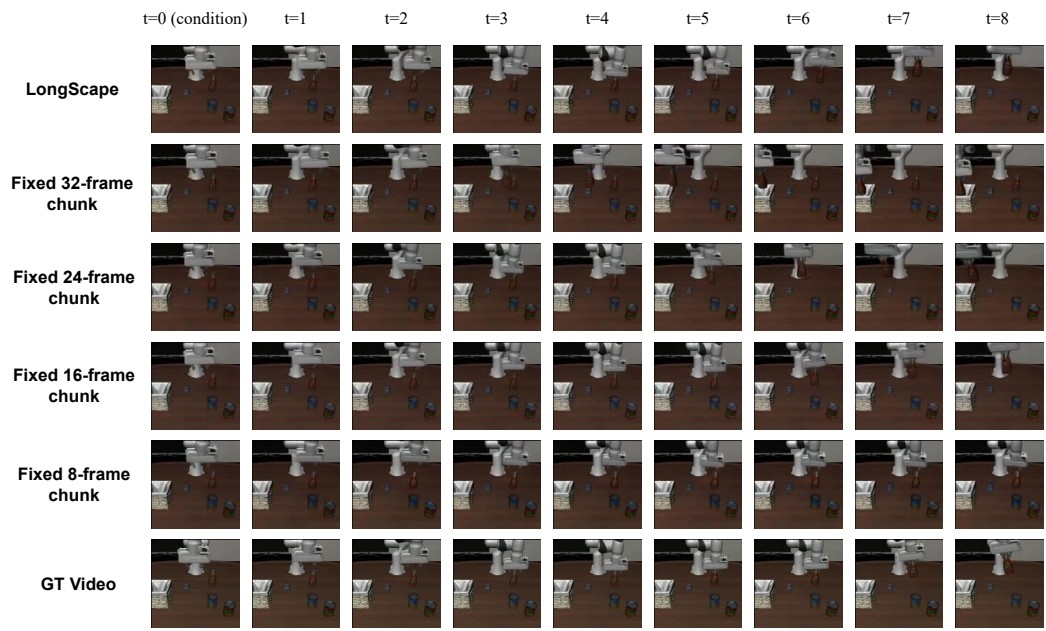

Figure 8: Visualization of the ablation study on LIBERO dataset. We compare LongScape with variants using only fixed-length chunks. These fixed-length approaches often exhibit discontinuous or physically implausible actions. In contrast, LongScape, by incorporating an adaptive expert mechanism, ensures action coherence and stable generation.

