# OpenReview forum: "LongScape: Advancing Long-Horizon Embodied World Models with Context-Aware MoE"
_ICLR.cc/2026/Conference — Submitted to ICLR 2026_

### Official Review · Reviewer_RiT6 · 2025-10-30

**Soundness:** 2
**Presentation:** 1
**Contribution:** 1
**Rating:** 2
**Confidence:** 4

**Summary:**

This paper presents an autoregressive video diffusion model with an internal structure designed using MoE. The experimental section mainly tests on robot datasets, with video lengths reaching up to 200 frames.

**Strengths:**

Based on the quantitative metrics provided by the authors, it is evident that the proposed method has certain advantages.

**Weaknesses:**

- (1) The paper claims to generate "Embodied World Models," but it seems that the model's input does not include actions. This seems more like a regular video generation model, giving the impression of leveraging "Embodied AI" without substantial connection.
- (2) The paper addresses the problem of Long-Horizon Video Generation, but it lacks comparison and discussion with many current SOTA long-video generation works, such as Diffusion Forcing, DFoT, FramePack, Vista, LCT, FAR, CausVid, Self-forcing, and many others. There are just too many relevant works to ignore. My concern is that the most advanced long-video generation models could achieve good results for embodied video generation, and the tricks like MoE used in this paper may ultimately not be the key design.
- (3) The video length in the experiments seems to be a maximum of 200 frames? Modern full-sequence video generation models, such as Wan, can generate at least 81 frames in a single pass. For sequences several hundred frames long, the computational cost using full-attention is entirely manageable. What I want to point out is: in the experiments in this paper, could the videos be generated directly using a full-sequence video generation model? It seems like there's no need for autoregressive generation at all. If the goal is to test the capability of autoregressive long-video generation, longer videos should be tested, with frame counts above 1000, for example.
- (4) The proposed method in this paper does not seem to be limited to robotic datasets. In fact, it can be applied to general long videos. If it’s only limited to robot datasets, this would diminish the value and inspiration of the proposed method.

**Questions:**

none

---

### Official Review · Reviewer_pDYr · 2025-11-01

**Soundness:** 3
**Presentation:** 3
**Contribution:** 3
**Rating:** 4
**Confidence:** 4

**Summary:**

This present LongScape, a hybrid generator that combines intra-chunk diffusion with inter-chunk autoregressive causality. The key is action-guided, variable-length chunking, which segments sequences by robotic action semantics so each chunk forms a coherent unit. A Context-aware Mixture-of-Experts (CMoE) then selects specialized experts per chunk, preserving detail and ensuring smooth transitions. Across benchmarks, LongScape delivers stable, consistent long-rollout generation.

**Strengths:**

1. It is a clear and practical idea that using action-prior variable-length chunking and context-aware MoE routing for semantically complete chunks and causally consistent rollouts.

2. It can clearly be reproduced, cause model sizes, training setup, and an anonymous repo are provided.

3. The paper is well-organized, with intuitive figures and sufficient experimental tables, and is easy to follow.

**Weaknesses:**

1. The evaluation metric leans on perceptual metrics or video quality metric, lacks temporal-consistency metrics, and simulator task successful rate for world-model utility.

2. Although only one expert is activated at inference, the paper does not report end-to-end latency, throughput, or VRAM comparisons against fixed-chunk baselines. Training cost remains high (multi-expert, multi-round), but the empirical gains are modest.

3. Missing strong baselines (e.g., MAGI-1) and ablation experiments to explain why using 8,16,32 frames but not others.

**Questions:**

Could you clarify which specific tasks were used for training and testing?

If the evaluation includes unseen tasks, does the proposed framework still generalize effectively in those settings?

Additionally, could you provide examples of longer-horizon, multi-view, or multi-object manipulation scenarios to further demonstrate the method’s robustness and generalization?

---

### Official Review · Reviewer_K8SU · 2025-11-06

**Soundness:** 3
**Presentation:** 3
**Contribution:** 2
**Rating:** 4
**Confidence:** 3

**Summary:**

1. The paper proposes a new method, LongScape, for long-horizon embodied world generation
2. LongScape encodes frame chunks in an autoregressive manner. Instead of tokenizing into discrete features, each chunk feature remains continuous and use conditional diffusion for generation
3. An action guided mechanism is used for partitioning. There are four types of chunks.
4. Each type of chunk has a specific expert. The mixture-of-experts and router are optimized with an additional expert id.

**Strengths:**

1. The paper is well written and easy to understand.
2. The proposed method is straightforward, elegantly incorporating the benefits of both diffusion and autoregressive generation.

**Weaknesses:**

1. The idea of generating long sequence embodied video is great. However, I am concerned with the baseline choices. The baselines are more general and not focus on embodied generation. Can you discuss more about this? Are there any embodied world model that are suitable for comparison, such as DreamGen?
2. The comparison in figure 4 is very obvious. Can you show video result or clearer version of qualitative comparison?

**Questions:**

1. Are you the first paper to address "Long Horizon" generation? Can you discuss more about this in the related work?
2. The action guided partition seems limited. Is the embodied world focus on gripper? Is the mechanism considers all the situation? The description about theta_d and alpha is not easy to understand? Can you describe in detail?
3. "MoE mechanism allows us to scale model capacity by a factor of K" Why?
4. I am not familiar with MoE. How is the MoE here different from general MoE? Your MoE seems to be deterministic (certain input-expert relationship)?

---

### Meta-Review · Area_Chair_j3pp · 2026-01-03

**Summary:**

This paper proposes a novel hybrid long video generation model. The key contribution is to segment the video sequence into chunks of variable length based on the action. Different DiTs are trained to for generating videos with different lengths and a context-based dynamic router is introduced to enable variable-length chunk generation. Reviewers acknowledge that the paper is well written and organized, it is reproducible.

Majors concerns of this paper are:
1. This paper claims to be "Embodied world models", however, the metrics and baselines are about general video generation.
2. Strong baselines are missing.

**Reviewer Concerns:**

1. Both reviewer K8SU and RiT6 mentioned that the paper claimed to be a "Embodied world models" however, the metrics and baselines are about general video generation. This is a valid concern.

2. Both reviewer pDYr and RiT6 have concerns about the baselines. Reviewer pDYr pointed out that MAGI-1 should be a strong baseline which is missing from the paper, RiT6 pointed out that the paper lacks comparison with many SOTA long-video generation works.

3. Both reviewer pDYr and RiT6 pointed out that more discussions/studies of the length of the generated videos and the choice of the length for each chunk are needed.

**Reviewer Scores:**

This paper get scores of 4, 4, 2. The authors did not post their rebuttal to address the concerns and questions of the reviewers, I tend to believe the reviewers will keep their scores.

---

### Decision · Program_Chairs · 2026-01-26

Reject